# Clinical Predictors for Procedural Stroke and Implications for Embolic Protection Devices during TAVR: Results from the Multicenter Transcatheter Aortic Valve Replacement In-Hospital Stroke (TASK) Study

**DOI:** 10.3390/jpm12071056

**Published:** 2022-06-28

**Authors:** Anat Berkovitch, Amit Segev, Elad Maor, Alexander Sedaghat, Ariel Finkelstein, Matteo Saccocci, Ran Kornowski, Azeem Latib, Jose M. De La Torre Hernandez, Lars Søndergaard, Darren Mylotte, Niels Van Royen, Azfar G. Zaman, Pierre Robert, Jan-Malte Sinning, Arie Steinvil, Francesco Maisano, Katia Orvin, Gianmarco Iannopollo, Dae-Hyun Lee, Ole De Backer, Federico Mercanti, Kees van der Wulp, Joy Shome, Didier Tchétché, Israel M. Barbash

**Affiliations:** 1Interventional Cardiology Unit, Leviev Heart and Vascular Center, Chaim Sheba Medical Center, Ramat Gan 5262000, Israel; anatberko@gmail.com (A.B.); amit.segev@sheba.health.gov.il (A.S.); elad.maor@sheba.health.gov.il (E.M.); 2Sackler School of Medicine, Tel-Aviv University, Tel Aviv 69978, Israel; afinkel@tlvmc.gov.il (A.F.); ran.kornowski@gmail.com (R.K.); ariksteinvil@gmail.com (A.S.); katiaorvin@gmail.com (K.O.); 3Heart Center Bonn, University Hospital Bonn, 53127 Bonn, Germany; alexander.sedaghat@ukbonn.de (A.S.); jan-malte.sinning@ukbonn.de (J.-M.S.); 4Division of Cardiology, Tel Aviv Medical Center, Tel Aviv 69651, Israel; 5Cardiovascular Surgery Department, University Hospital of Zurich, CH-8091 Zurich, Switzerland; dr.saccocci@gmail.com (M.S.); francesco.maisano@usz.ch (F.M.); 6Division of Cardiology, Rabin Medical Center, Petach-Tikva 49100, Israel; 7Department of Cardiology, Montefiore Medical Center, New York, NY 10467, USA; alatib@gmail.com; 8Interventional Cardiology Department, Hospital Universitario Marques de Valdecilla, 39008 Santander, Spain; he1thj@humv.es; 9The Heart Center, Rigshospitalet, Blegdamsvej 9, 2100 Copenhagen, Denmark; lars.soendergaard.01@regionh.dk (L.S.); daehyun.lee@scsalud.es (D.-H.L.); 10University Hospital and SAOLTA Health Care Group, National University of Ireland, H91 TK33 Galway, Ireland; darrenmylotte@gmail.com (D.M.); ole.debacker@gmail.com (O.D.B.); 11Department of Cardiology, Radboud University Medical Center, Postbus 9101, 6500 HB Nijmegen, The Netherlands; niels.vanroyen@radboudumc.nl; 12Cardiology, Freeman Hospital and Institute of Cellular Medicine, Newcastle University, Newcastle upon Tyne NE1 7RU, UK; azfar.zaman@nhs.net (A.G.Z.); kees.vanderwulp@radboudumc.nl (K.v.d.W.); 13Department of Interventional Cardiology, Clinique Pasteur, 31300 Toulouse, France; pierrecardio@gmail.com (P.R.); joy.s.shome@gmail.com (J.S.); 14Interventional Cardiology Unit, San Raffaele Hospital, 10090 Milan, Italy; gianmarco.iannopollo@ausl.bologna.it (G.I.); dtchetche@clinique-pasteur.com (D.T.); 15Division of Cardiology, University of Rome Tor Vergata, 00173 Rome, Italy; federico.mercanti@yahoo.it

**Keywords:** transcatheter aortic valve replacement, aortic stenosis, stroke

## Abstract

**Background**: Data to support the routine use of embolic protection devices for stroke prevention during transcatheter aortic valve replacement (TAVR) are controversial. Identifying patients at high risk for peri-procedural cerebrovascular events may facilitate effective patient selection for embolic protection devices during TAVR. **Aim**: To generate a risk score model for stratifying TAVR patients according to peri-procedural cerebrovascular events risk. **Methods and results:** A total of 8779 TAVR patients from 12 centers worldwide were included. Peri-procedural cerebrovascular events were defined as an ischemic stroke or a transient ischemic attack occurring ≤24 h from TAVR. The peri-procedural cerebrovascular events rate was 1.4% (*n* = 127), which was independently associated with 1-year mortality (hazards ratio (HR) 1.78, 95% confidence interval (CI) 1.06–2.98, *p* < 0.028). The TASK risk score parameters were history of stroke, use of a non-balloon expandable valve, chronic kidney disease, and peripheral vascular disease, and each parameter was assigned one point. Each one-point increment was associated with a significant increase in peri-procedural cerebrovascular events risk (OR 1.96, 95% CI 1.56–2.45, *p* < 0.001). The TASK score was dichotomized into very-low, low, intermediate, and high (0, 1, 2, 3–4 points, respectively). The high-risk TASK score group (OR 5.4, 95% CI 2.06–14.16, *p* = 0.001) was associated with a significantly higher risk of peri-procedural cerebrovascular events compared with the low TASK score group. **Conclusions:** The proposed novel TASK risk score may assist in the pre-procedural risk stratification of TAVR patients for peri-procedural cerebrovascular events.

## 1. Introduction

The application of transcatheter aortic valve replacement (TAVR) is expanding to include lower-risk patients. Although recent studies have shown that rates of stroke after TAVR are low (0.6–3.4%) [1,2], the significant morbidity and mortality associated with it render it a priority target for preventative measures [3]. Cerebrovascular protection during TAVR has emerged as a new approach for the prevention of peri-procedural strokes. The devices are designed to capture embolic debris released during the TAVR procedure, thereby providing protection during the procedure itself as well as in the period immediately following it. However, the lack of compelling evidence for clinical stroke event reduction, the high cost, and the additional procedural complexity of device utilization have limited the widespread adoption of such technologies.

Tools to identify TAVR candidates at high risk for peri-procedural stroke may be valuable in selecting patients who require cerebrovascular protection during TAVR. Furthermore, such stratification tools may facilitate the future evaluation of the clinical efficacy of embolic protection devices among patients with different levels of risk for peri-procedural cardiovascular events. Earlier studies had evaluated predictors for post-procedural stroke [4,5,6,7], but peri-procedural cerebrovascular events represent different pathophysiology factors than those of cerebrovascular events that occur later during follow-up (i.e., >24 h). Thus, such predictors may no longer be pertinent to predicting peri-procedural events and may not be relevant for the clinical assessment of patients prior to the procedure. 

The aims of the current study are to identify predictors and to design a scoring system to stratify the risk of a TAVR-associated cerebrovascular event in order to assist in identifying patients at high risk for peri-procedural strokes who may benefit from embolic protection devices during TAVR.

## 2. Methods

The data that support the findings of this study are available from the corresponding author upon reasonable request.

The TASK (*Transcatheter Aortic valve replacement in-hoSpital stroKe*) study included consecutive TAVR patients from 12 high-volume TAVR centers in Europe and the Middle East that contributed their data (Appendix A). All the study patients had undergone TAVR procedures between 2007 and 2018 after careful evaluation by each institutional heart team. All patients undergoing the TAVR procedure during the designated period were included in the analysis. Patients who underwent TAVR via a non-transfemoral approach and those in whom an embolic protection device was utilized were excluded. All centers used heparin during the TAVR procedure, with a target activated clotting time of >250 s.

The participating centers were required to fill out a case report form designed specifically for this study, which included information on demographics, past medical history, and medications. Low body weight was defined as a body mass index of ≤25 kg/m^2^ and ischemic heart disease as any prior acute coronary syndrome, coronary intervention, or bypass surgery. Chronic kidney disease was defined as a glomerular filtration rate of <60 mL/min/1.73 m^2^ based on the Modification of Diet in Renal Disease equation and peripheral vascular disease as any documented atherosclerotic disease in the carotid arteries, renal arteries, or any peripheral arteries. Outcome data were collected according to the Valve Academic Research Consortium-2 definitions [8] and included post-procedural complications with a special focus on ischemic stroke, transient ischemic attack, and hemorrhagic stroke, as well as the timing of their occurrence. Any inconsistencies in the data were resolved directly with local investigators and on-site data monitoring. All patients gave written informed consent to undergo a transcatheter aortic valve procedure. The inclusion of patients was approved in each center by a local ethics committee.

The primary endpoint of peri-procedural cerebrovascular events was defined as the composite of peri-procedural ischemic stroke or transient ischemic attack occurring earlier than 24 h post-procedure. Cerebrovascular events were defined according to the Valve Academic Research Consortium-2 definitions [8] and categorized as transient ischemic attack or ischemic stroke. “Transient ischemic attack” was defined as a sensorimotor deficit that lasted 24 h or less without associated evidence of cerebral infarction in imaging studies. “Ischemic stroke” was defined as an acute neurological dysfunction lasting at least 24 h with or without evidence of infarction in imaging studies. TAVR patients for whom there was no information on a stroke event or the timing of a stroke with respect to the TAVR procedure were excluded from the analysis.

## 3. Statistical Analysis

Continuous data were compared with Student’s t-test and one-way ANOVA. Categorical data were compared with the chi-square test or the Fisher exact test. To assess the prognostic impact of a peri-procedural cerebrovascular event, a multivariate Cox regression analysis adjusted for age, gender, ischemic heart disease, hypertension, diabetes mellitus, peripheral vascular disease, chronic kidney disease, ejection fraction, and previous stroke was performed. Odds ratios (ORs) are reported as absolute values and 95% confidence intervals (CIs). 

Candidate parameters that were considered for the derivation of the TASK score included exclusively the pre-procedural parameters that are available to the physician prior to the procedure in order to identify pre-procedural predictors for stroke. 

Univariate logistic binary regression modeling was used to evaluate the ORs for peri-procedural cerebrovascular events. The final components selected for inclusion in the TASK score were derived using a resampling-based procedure. The relative impact of each of the pre-procedural parameters in predicting stroke was determined and ranked. The frequency of each candidate parameter in a final model derived from 1 of 1000 bootstrapped samples served as an indication of the importance of that parameter (Appendix B). 

Parameters identified as most important were included in a multivariable logistic regression and were inspected to confirm whether equal weight can be given to each parameter so that a TASK score can be created by counting the number of present parameters for an individual. The C statistic was used to assess the performance of the multivariable model (Appendix B). 

The TASK score was designed by assigning a single point to each significant factor. The TASK score was further dichotomized into four risk groups: very-low, low, intermediate, and high (0, 1, 2, 3–4 points, respectively). The prognostic value of the TASK score was assessed using a receiver operating characteristics analysis, producing an area under the curve with 95% CIs. The predicted and observed incidence of peri-procedural cerebrovascular events was compared using the Hosmer–Lemeshow test to assess the goodness-of-fit of the model. The model was regarded as having no goodness-of-fit if *p *< 0.05. In order to assess the possible effects of a procedural learning curve on the incidence of peri-procedural cerebrovascular events and on the TASK risk score, we defined “early” and “late” periods according to procedures performed before or after the median procedure date, respectively, and performed an interaction analysis between these two time periods.

Given the low rates of peri-procedural cerebrovascular events, a separate validation cohort was not available. 

Statistical significance was accepted for a two-sided *p* < 0.05. The statistical analyses were performed with IBM SPSS version 25.0 (Chicago, IL, USA) and with SAS Enterprise Guide version 7.1.

## 4. Results

A total of 8779 patients underwent TAVR during the study period, of whom 4546 (52%) were females. The median age of the cohort was 82 years (IQR 79–86 years). The baseline characteristics of the study population are summarized in Table 1. In total, 100% of patients in the cerebrovascular group had an event, while only 0.9% of the control group had an event. The 0.9% are events that occurred after the first 24 h of the procedure. The procedure was performed under conscious sedation in the majority of patients (69%). A self-expandable valve was utilized most frequently (57%), followed by a balloon expandable valve (37%) and a mechanically expandable valve (6%). Balloon pre-dilatation was performed in one-half of the cases. 

The in-hospital adverse events rate was low, with the total in-hospital cerebrovascular event rate of 2.3%, most of which were ischemic strokes (86%), followed by transient ischemic attacks (14%), and hemorrhagic strokes (0.1%). Life-threatening or major bleeding events occurred in 7% of the patients, and the in-hospital mortality rate was 1.2% (Table 1). 

### 4.1. Peri-Procedural Cerebrovascular Events

Stroke events within 72 h occurred in 145 patients (1.7%): 127 events occurred <24 h post-procedure (1.4%), 12 events during the second post-procedural day (0.14%), and only 6 events on the third post-procedural day (0.07%). 

Evaluating patients with peri-procedural cerebrovascular events <24 h post-TAVR, the univariate binary regression analysis identified several parameters associated with peri-procedural cerebrovascular events: chronic kidney disease (OR 2.16, 95% CI 1.36–3.43), the use of non-balloon expandable valves (OR 1.89, 95% CI 1.21–2.94), peripheral vascular disease (OR 1.76, 95% CI 1.19–2.62), and a previous stroke event (OR 1.60, 95% CI 0.91–2.8) (Table 1). Of note, common risk factors for stroke, such as older age, diabetes, and atrial fibrillation, were not associated with peri-procedural cerebrovascular events (Table 1).

A multivariate binary analysis comprised of 6131 patients with complete baseline information identified four independent predictors of peri-procedural cerebrovascular events: a previous stroke (OR 1.84, 95% CI 1.01–3.34), the use of non-balloon expandable valves (OR 2.06, 95% CI 1.29–3.30), chronic kidney disease (OR 2.14, 95% CI 1.31–3.51), and peripheral vascular disease (OR 1.82, 95% CI 1.19–2.80) (Figure 1, Appendix A). These findings were substantiated by bootstrap analysis (Appendix A). The final components selected for inclusion in the TASK score were derived using a resampling-based procedure.

### 4.2. TASK Score Derivation and Validation

The TASK score was derived by assigning a single point to each of the selected components, and it ranged between 0–4, including previous stroke, the use of non-balloon expandable valves, chronic kidney disease, and peripheral vascular disease. All four variables had a statistically equivalent impact on stroke risk. The event rates were 0.7%, 0.8%, 2.1%, 3.4%, and 7.8% for each increment in the TASK score points of 0–4, respectively (Figure 2). Each one-point increment in the TASK risk score was associated with a significant increase in the risk of peri-procedural cerebrovascular events (OR 1.96, CI 1.56–2.45, *p* < 0.001) with a C statistic of 0.65 ± 0.03 (95% CI 0.60–0.71; Figure 3). The Hosmer–Lemeshow goodness-of-fit test demonstrated that the TASK model was well-calibrated, with a non-significant *p*-value of 0.84. Cross-validation of the TASK score using resampling-based metrics confirmed the robustness of the derivation model with a cross-validated C statistic of 0.641 and an optimism-corrected C statistic of 0.0038 (Appendix A).

An interaction analysis demonstrated that the association between the TASK score and peri-procedural cerebrovascular events was consistent in both the early and late study periods (HR 2.12 (1.61–4.79) for the early period vs. 1.51 (1.12–2.05) for the late period, *p* for interaction 0.106). 

The TASK score was dichotomized into four mutually exclusive groups according to the predicted risk for peri-procedural cerebrovascular events, i.e., very-low, low, intermediate, and high (0, 1, 2, 3–4 points, respectively). Patients assigned to the low-risk group (one point) had a non-significant increase in the risk of peri-procedural cerebrovascular events compared with the very-low TASK score group (0 points) (OR 1.14 95% CI 0.42–3.06). However, patients in the intermediate-risk group (2 points) had a significant increase in the risk of peri-procedural cerebrovascular events compared with the very-low TASK score group (0 points) (OR 2.9, 95% CI 1.16–7.37) (Table 2). Moreover, patients in the high (3–4 points) TASK score group showed a significantly higher risk of peri-procedural cerebrovascular events compared with the very-low TASK score group, with an OR of 5.4 for the high TASK score (95% CI 2.06–14.16, *p* = 0.001) (Table 2). 

### 4.3. Prognostic Value of Peri-Procedural Cerebrovascular Events

A total of 1166 (14%) patients died within one year after undergoing a TAVR procedure. The Kaplan–Meier survival analysis demonstrated that patients who sustained peri-procedural cerebrovascular events had significantly higher 1-year mortality compared to patients who did not (*p* < 0.001) (Figure 4). The multivariate Cox regression analysis adjusted for age, gender, ischemic heart disease, hypertension, diabetes mellitus, peripheral vascular disease, chronic kidney disease, ejection fraction, and previous stroke found that peri-procedural cerebrovascular events were an independent risk factor for mortality, with an increased 1-year mortality risk compared to patients who did not undergo a stroke (HR 1.78, 95% CI 1.06–2.98, *p* < 0.028) (Appendix A).

## 5. Discussion

The current study represents the first attempt to perform a multicenter, all-comer analysis to create a clinically relevant score for stratifying patients at high risk for peri-procedural cerebrovascular events during or after a TAVR procedure. The importance of the TASK score stems from the fact that although the rates of peri-procedural cerebrovascular events are low (1.4%), they are associated with increased mortality. The proposed TASK score utilizes exclusively the parameters that are known prior to the procedure to stratify patients into four distinct risk groups for peri-procedural cerebrovascular events. The TASK score may, therefore, be utilized during the pre-procedural evaluation process as guidance for the use of embolic protection devices in high-risk patients and may thus serve as a practical tool to reduce the risk of peri-procedural cerebrovascular events during TAVR or shortly thereafter.

The findings in this study, as well as those of prior studies [4,9,10], indicate that stroke after TAVR is an independent predictor of increased mortality and morbidity, such that patients who sustain an acute cerebrovascular event have a 6.5-fold increased risk of 30-day mortality [9]. Considering the expansion of TAVR to low-surgical-risk patients and younger populations, tools to identify patients at high risk for peri-procedural cerebrovascular events are essential in order to decrease the risk of disabling stroke. 

A number of devices have been developed to prevent cerebrovascular embolization during the TAVR procedure [10]. Seeger and colleagues performed a retrospective analysis on the use of an embolic protection device in TAVR procedures and showed a decrease in stroke rates related to its utilization [11]. However, other randomized control trials and observational studies, including two separate meta-analyses [12,13] that evaluated several major randomized control trials [10,14,15,16,17,18], did not demonstrate a statistically significant decrease in clinically overt stroke rates with the use of embolic protection devices. Several studies have shown that embolic protection devices may decrease the formation of new brain lesions, as demonstrated in diffusion-weighted magnetic resonance imaging; however, none were associated with a reduction in clinical stroke events [10]. Therefore, embolic protection devices are used in a small minority (13%) of TAVR cases in the United States [19]. The available data, however, may support the need for a pre-procedural risk assessment tool and the use of protection devices in selected high-risk patients. The TASK score was developed to provide a simple tool to identify the patients at high risk for peri-procedural stroke, in which cerebrovascular protection may yield a high risk–benefit ratio. Furthermore, the TASK score may be utilized in the design of future clinical studies that evaluate the safety and efficacy of new embolic protection devices and contribute to the identification of a specific group of patients who will benefit most from such protection during the TAVR procedure. 

Prior studies that evaluated clinical predictors of stroke after TAVR had assessed all stroke events, including those occurring late (>48 h following the procedure) [5,7]. Those studies identified parameters such as female gender, acute renal failure, chronic obstructive pulmonary disease, and low body weight as predictive factors [5,7]. Of note, those proposed predictors of stroke were not evaluated according to the timing of the event but rather to the occurrence of an event at any time during follow-up. However, as opposed to late events, the initial 24 hours after TAVR represent a unique period of vulnerability for acute cerebrovascular events. Stroke events occurring during this period account for the majority of cases [20], and they may have different pathophysiology, mainly attributable to procedural factors that are embolic in nature [21] and that result from device manipulation within the aortic arch and the calcified aortic valve. Such manipulations may lead to the dislodgement of micro-particles and the subsequent embolization of debris from an atheroma or from the valve itself [5]. Thus, the predictors of stroke following TAVR that were reported in earlier publications may not necessarily be relevant for predicting peri-procedural events.

Despite the increased risk of stroke and the different pathophysiology of peri-procedural stroke, few studies have explored the predictors of acute stroke by identifying technical procedural elements such as the balloon post-dilatation, the number of implantation attempts, and valve embolization [4,20]. However, none of these factors can be accounted for during a pre-procedural evaluation, and a more practical tool is needed in order to stratify patients according to the risk of acute stroke. Thus, the present analysis is focused upon pre-procedural parameters with potential impact on the likelihood of peri-procedural cerebrovascular events. We identified a history of stroke, the use of non-balloon expandable valves, chronic kidney disease, and peripheral vascular disease as predictors of peri-procedural stroke, and these parameters were incorporated into the new TASK score. One shared characteristic of all these factors is that they are available to the physician prior to the procedure, thereby enabling pre-procedural risk stratification. Importantly, the presence of all predictors (a TASK score = 4) was associated with a more-than-11-fold increased risk of peri-procedural cerebrovascular events compared with patients without those factors. The majority of the identified predictors are well-established predictors for stroke at any time after the procedure, i.e., peripheral artery disease, chronic kidney disease, and history of stroke. These predictors may increase the risk for stroke at any time but also during the peri-procedural period. The emergence of the use of non-balloon expandable transcatheter heart valves as a predictor for peri-procedural stroke may be related to increased rates of post-dilatation [22] or other maneuvers in the aortic arch. Of note, traditional risk factors for stroke (e.g., atrial fibrillation) were not associated with peri-procedural events, probably reflecting the homogeneity of age of this elderly group of patients and the different pathophysiologies involved in peri-procedural versus late stroke events. 

## 6. Limitations

The present study has several limitations. The TASK score was designed to identify patients at high risk for peri-procedural cerebrovascular events in order to help clinicians identify, prior to the procedure, those patients who may benefit from the implementation of cerebrovascular protection. Therefore, procedural factors that may have a significant impact on the risk of acute stroke were not included in the present analysis despite the fact that they might influence the risk of stroke. Given the retrospective nature of the study and the variability in clinical practice among participating centers, there was no standardization in the evaluation of the patients who sustained a cerebrovascular event, no mandatory neurological evaluation by a neurologist, and no routine assessment of the modified Rankin scale. Additionally, not all patients underwent head computed tomography or brain magnetic resonance imaging. All the participating sites did, however, use the VARC-2 criteria to define cerebrovascular events. Finally, information on pre-procedural CTs was not available for all patients. Therefore, calcification severity or other CT parameters were not integrated into the model. 

## 7. Conclusions

In conclusion, the TASK score represents a possible stratification tool for TAVR candidates according to the risk of cerebrovascular events during or immediately after the procedure. The score is comprised of clinical parameters readily available prior to undertaking the procedure. The utilization of the TASK score may serve as an additional tool for clinicians who are considering providing cerebrovascular protection during a TAVR procedure to a given patient.

## Figures and Tables

**Figure 1 jpm-12-01056-f001:**
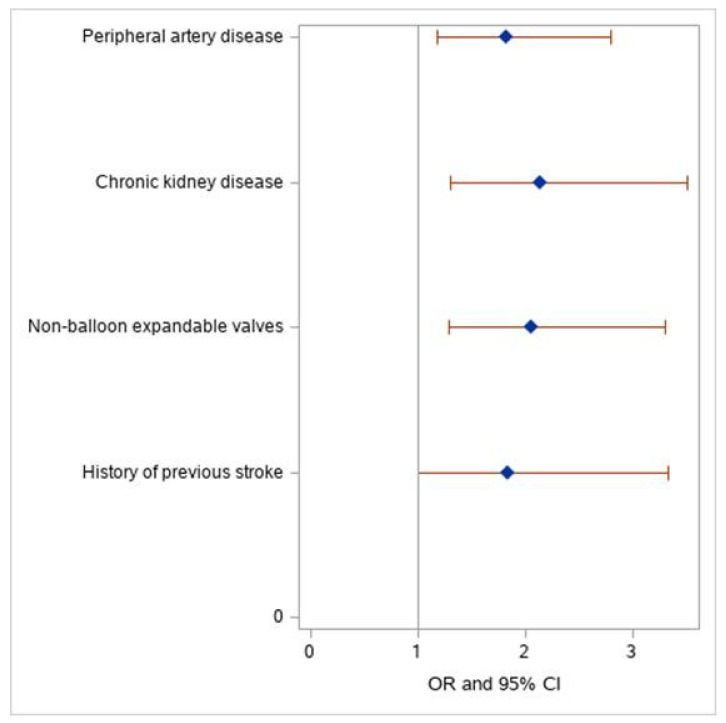
Odds ratio with 95% confidence limits. Multivariate Cox regression analysis for independent predictors for peri-procedural cerebrovascular events. Forest plot graph demonstrating the odds ratio of the TASK score components for peri-procedural cerebrovascular events.

**Figure 2 jpm-12-01056-f002:**
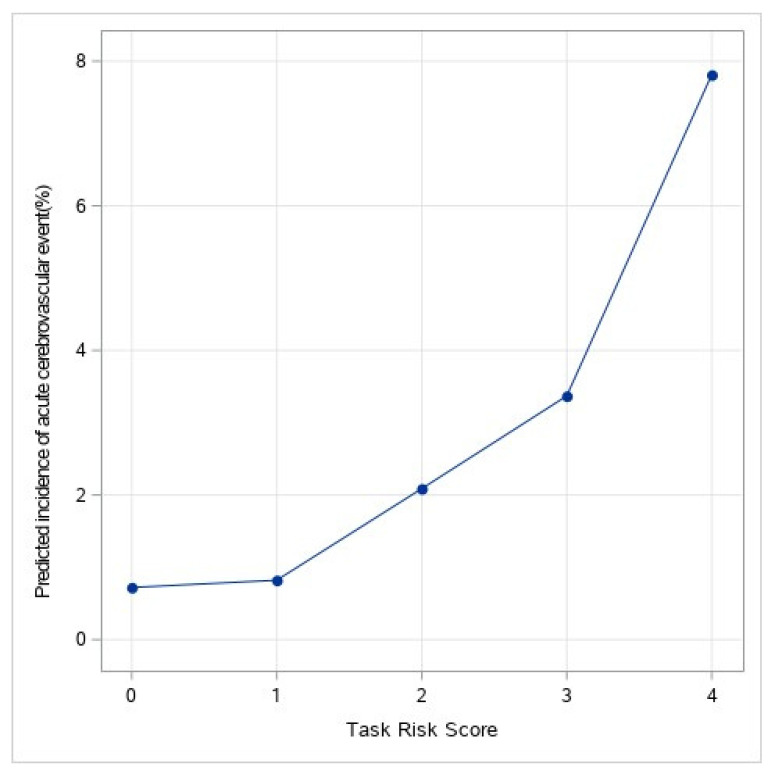
Predicted incidence of acute cerebrovascular event (%). Predicted incidence of peri-procedural cerebrovascular event (%) according to the TASK score.

**Figure 3 jpm-12-01056-f003:**
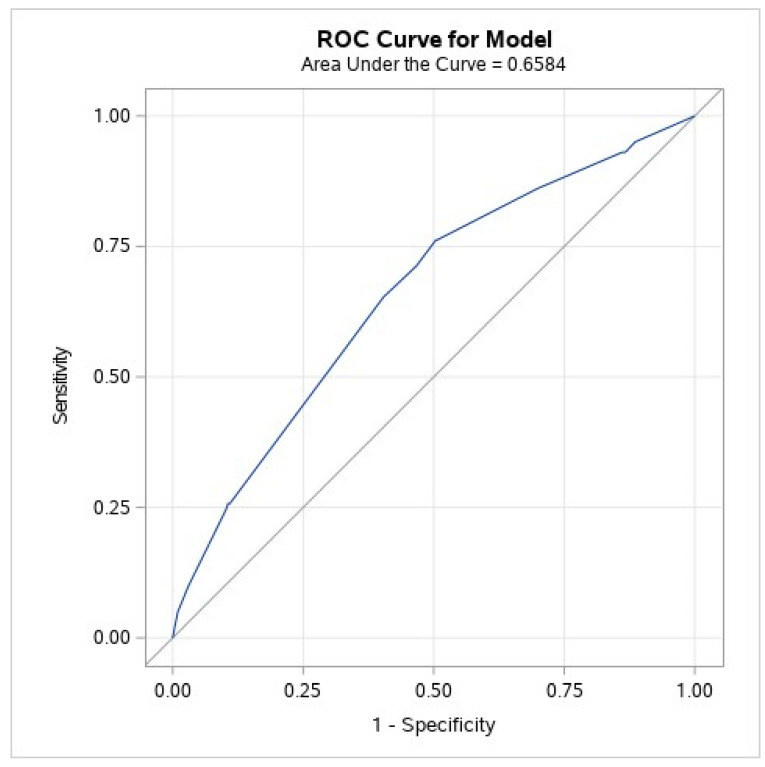
ROC curve (original data). Receiver operator curve. The value of the TASK score was analyzed by receiver operator curve analysis for the prediction of peri-procedural cerebrovascular events (AUC: 0.65, 95% confidence interval 0.60–0.71).

**Figure 4 jpm-12-01056-f004:**
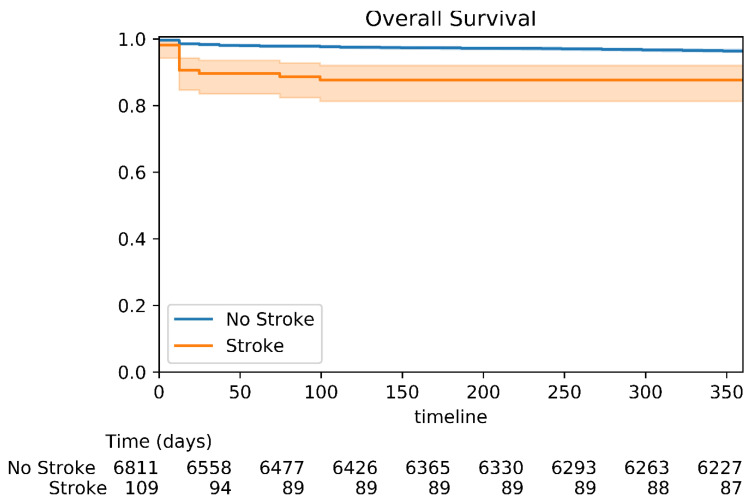
Kaplan–Meier survival analysis for 1-year mortality. Kaplan–Meier survival analysis for 1-year mortality showing the probability of mortality at the 1-year follow-up according to the peri-procedural status. *P*-value log rank <0.001.

**Figure A1 jpm-12-01056-f0A1:**
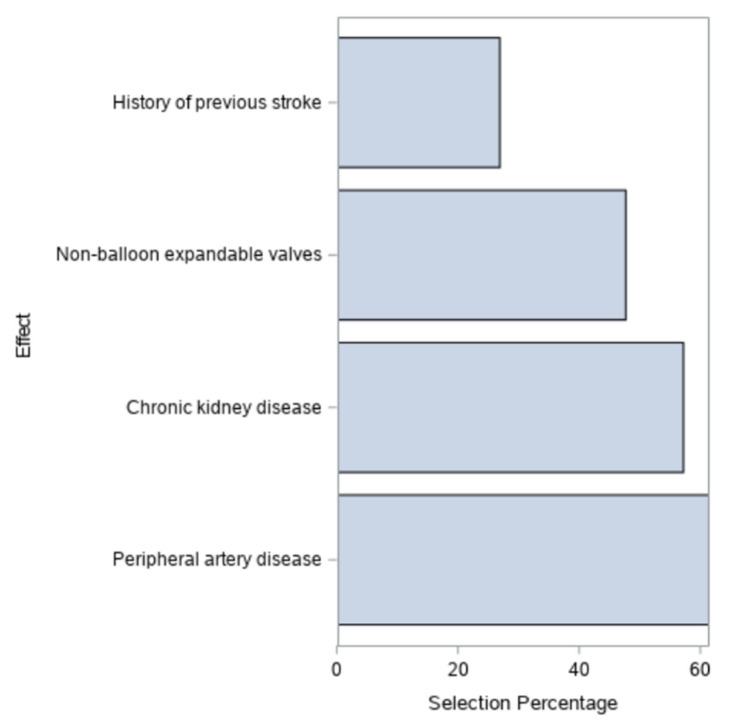
GLMSELECT Effect Selection in at Least 20% of the samples for Post CVA in 24 hours.

**Table 1 jpm-12-01056-t001:** Demographic, clinical, and procedural characteristics of the patients according to peri-procedural cerebrovascular events.

Variable	Entire Cohort	Peri-Procedural Cerebrovascular Event		Odds Ratio	Confidence Interval	*p*-Value
	N = 8779	Yes N = 127	No N = 8652			
Baseline characteristics						
Age (mean ± SD)	82 ± 6.6	82.1 ± 6.8	83.1 ± 6.5	1.02	0.99–1.05	0.12
Female gender (%)	4546 (52)	72 (57)	4474 (52)	1.22	0.86–1.74	0.27
Low body weight * (%)	3414 (40)	65 (52)	3349 (40)	1.60	1.12–2.28	0.009
Ischemic heart disease (%)	2697 (31)	39 (31)	2658 (31)	0.95	0.67–1.44	0.95
Chronic kidney disease ** (%)	5458 (68)	101 (82)	5357 (68)	2.16	1.36–3.43	<0.001
Stroke history (%)	620 (7)	14 (11)	606 (7)	1.60	0.91–2.8	0.10
Diabetes mellitus (%)	2586 (30)	35 (28)	2551 (30)	0.9	0.61–1.33	0.60
Hypertension (%)	6088 (80)	79 (81)	6009 (80)	1.06	0.64–1.78	0.81
Atrial fibrillation (%)	2607 (32)	35 (29)	2572 (32)	0.85	0.57–1.26	0.43
Chronic obstructive pulmonary disease (%)	1296 (17)	17 (16)	1279 (17)	0.97	0.58–1.65	0.92
Peripheral vascular disease (%)	1462 (19)	35 (29)	1427 (19)	1.76	1.19–2.62	0.005
Baseline medications						
Aspirin (%)	5646 (64)	66 (68)	4478 (62)	1.29	0.84–1.98	0.24
P2Y12 inhibitor (%)	1966 (30)	33 (35)	1933 (30)	1.26	0.82–1.93	0.28
Oral anti-coagulant (%)	1689 (26)	19 (20)	1670 (26)	0.72	0.44–1.20	0.21
Baseline Echocardiography						
AVA (cm^2^) (mean ± SD)	0.73 ± 0.2	0.74 ± 0.2	0.72 ± 0.2	0.83	0.34–2.03	0.68
Ejection fraction (mean ± SD)	54 ± 12	54 ± 12	54 ± 12	1.00	0.99–1.01	0.75
Mean gradient (mmHg) (mean ± SD)	45 ± 16	45 ± 14	45 ± 16	0.99	0.99–1.01	0.84
Procedural data						
Conscious sedation (%)	2675 (31)	37 (29)	2638 (31)	0.92	0.63–1.35	0.67
Self-expandable valve	4516 (52)	71 (56)	4445 (52)	1.35	0.91–2.00	0.136
Balloon expandable valve	2878 (37)	26 (24)	2852 (37)	0.53	0.34–0.83	0.005
Mechanical expandable valve	451 (6)	13 (12)	438 (6)	2.23	1.24–4.01	0.001
Balloon pre-dilatation	4358 (50)	68 (53)	4290 (50)	1.14	0.80–1.62	0.46
Balloon post-dilatation	1414 (19)	22 (19)	1392 (19)	0.94	0.62–1.57	0.98
In-hospital events						
Myocardial infarction (%)	41 (0.5)	3 (2.6)	38 (0.5)	5.4	1.65–17.8	0.005
Any cerebrovascular event (%)	203 (2.3)	127 (100)	77 (0.9)	-	-	<0.001
24 hr cardiovascular event (%)	127 (1.4)	127 (100)	0 (0)	-	-	<0.001
New atrial fibrillation (%)	600 (8)	9 (8)	591 (8)	1.03	0.52–2.05	0.934
Life-threatening/major bleeding (%)	606 (7)	12 (10)	594 (7)	1.42	0.78–2.59	0.251
In-hospital mortality (%)	867 (1.2)	10 (7.9)	97 (1.1)	7.54	3.83–14.82	<0.001

***** Body mass index ≤25 kg/m^2^; ** glomerular filtration rate <60 mL/min/1.73 m^2.^

**Table 2 jpm-12-01056-t002:** TASK risk score *.

	Observed Stroke Incidence (%)	Odds Ratio	Confidence Interval	*p*-Value
Very-low risk = 0 points (*n* = 692)	0.7	1.00	-	-
Low risk = 1 point (*n* = 2310)	0.8	1.14	0.42–3.06	0.79
Intermediate risk = 2 points (*n* = 2442)	2.1	2.93	1.16–7.37	0.022
High risk = 3–4 points (*n* = 687)	3.8	5.40	2.06–14.16	0.001

* The TASK score includes peripheral vascular disease, valve type, history of stroke, and chronic kidney disease.

## Data Availability

The data presented in this study are available on request from the corresponding author. The data are not publicly available due to privacy issues.

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
