# Peer review of "Clinical Predictors for Procedural Stroke and Implications for Embolic Protection Devices during TAVR: Results from the Multicenter Transcatheter Aortic Valve Replacement In-Hospital Stroke (TASK) Study"

_jpm, 2022, doi:10.3390/jpm12071056_

Round 1

Reviewer 1 Report

Anat Berkovitch et al. present a multicentric retrospective study to find a score to help predict the risk of cerebral stroke in patients undergoing TAVR. The score they present is based on pre-operative variables in order to give the possibility -among other things- to select patients who might benefit from embolic protection devices. The study is well conducted. I have only some minor comments.

-English should be reviewed throughout the manuscript (ie: the first sentence of the methods section of the abstract is poorly written, the abbreviation “CVE” should be plural and there is no concordance between subject and verb –for example in the condensed abstract: “We aimed to generate a risk score model which stratify transcatheter aortic valve replacement”/methods section: “The predicted and observed incidence of peri-procedural cerebrovascular events was compared”).

-In table 1, the variable “Cerebrovascular events” reported 100% in the group who had cerebrovascular events, and 77 (0.9%) in the group without cerobrovacular events. Could the authors explain it?

-Could the authors make a different paragraph for the Conclusions? They are together with the limitations.

Reviewer 2 Report

Thanks for the opportunity to review this study. The creation of a score to predict risk of stroke during TAVR is an interesting task. However, I am not so sure about how applicable in the clinical realm this score would actually be.  The study is methodologically well-conceived and clear. The sample size is fantastic, and the statistical analysis seems rigorous.

Please find below some recommendations that may improve the quality of this work:

As proposed by the authors, the scoring system recognizes 4 different variables: peripheral vascular disease, valve type, history of stroke, and chronic kidney disease:

1. Why are all the variables scored with 1 point? Which of those 4 variables is the most significant one (statistically wise)?

2. Please provide a better discussion of the pathophysiological mechanism by which those 4 risk factors are associated with an increased risk of stroke during TAVR (e.g. why having a certain type of valve is associated with an increased risk of stroke?)

3. My main concern: are not all TAVR procedures currently performed with a distal balloon to avoid embolic strokes? How would the score proposed in this paper actually change the current TAVR practice?

4. Some of the conclusions are not fully justified by the current results. Please soften the language of the Conclusions section.
